# Thermochromic Fibers via Electrospinning

**DOI:** 10.3390/polym12040842

**Published:** 2020-04-06

**Authors:** Jimmy Nguyen, Ratib M. Stwodah, Christopher L. Vasey, Briget E. Rabatin, Benjamin Atherton, Paola A. D’Angelo, Kathleen W. Swana, Christina Tang

**Affiliations:** 1Chemical and Life Science Engineering, Virginia Commonwealth University, Richmond, VA 23284, USA; nguyenj28@vcu.edu (J.N.); stwodahrm@vcu.edu (R.M.S.); vaseycl@vcu.edu (C.L.V.); rabatinb@vcu.edu (B.E.R.); athertonbe@vcu.edu (B.A.); 2U.S. Army Combat Capabilities Development Command Soldier Center, Natick, MA 01760, USAkathleen.w.swana.civ@mail.mil (K.W.S.)

**Keywords:** electrospinning, fiber, coaxial, liquid crystal, thermochromic, nonwoven

## Abstract

Cholesteryl ester liquid crystals exhibit thermochromic properties related to the existence of a twisted nematic phase. We formulate ternary mixtures of cholesteryl benzoate (CB), cholesteryl pelargonate (CP), and cholesteryl oleyl carbonate (COC) to achieve thermochromic behavior. We aim to achieve thermochromic fibers by incorporating the liquid crystal formulations into electrospun fibers. Two methods of incorporating the liquid crystal (LC) are compared: (1) blend electrospinning and (2) coaxial electrospinning using the same solvent system for the liquid crystal. For blend electrospinning, intermolecular interactions seem to be important in facilitating fiber formation since addition of LC can suppress bead formation. Coaxial electrospinning produces fibers with higher nominal fiber production rates (g/hr) and with higher nominal LC content in the fiber (wt. LC/wt. polymer assuming all of the solvent evaporates) but larger fiber size distributions as quantified by the coefficient of variation in fiber diameter than blend electrospinning with a single nozzle. Importantly, our proof-of-concept experiments demonstrate that coaxially electrospinning with LC and solvent in the core preserves the thermochromic properties of the LC so that thermochromic fibers are achieved.

## 1. Introduction

Due to growing interest in wearable technology, there is a need for new functional soft, lightweight, elastic materials that facilitate development of wearable devices [1,2,3,4]. Functional fibers that enable sensing without relying on electrical power are of particular interest [2]. For example, thermochromic fibers that change color as a function of temperature have been used as wearable sensors [4,5].

Cholesteric liquid crystals are a class of unique soft materials with thermochromic properties arising from their molecular structure [6,7]. Specifically, cholesteric liquid crystals form a twisted nematic phase [8,9] with a helical structure that have temperature-dependent pitch length [10]. A decrease in temperature causes untwisting of the helical structure leading to an increase in the pitch length [8,9]. At high temperatures (i.e., above the mesophase transition temperature), the isotropic liquid appears clear [8,9]. Upon cooling, the liquid crystal phase reflects visible light due to Bragg reflection [8,9] leading to strong iridescent color [10]. As the temperature decreases, the liquid crystal first reflects blue light (λ = 450 nm) at relatively short pitch lengths [8,9]. With further decreases in temperature, the wavelength reflected shifts to the red end of the spectrum (λ = 760 nm) due to the increase in pitch length [8,9]. Eventually the liquid crystal phase appears opaque as the resulting increase in pitch reflects light outside the visible wavelength range. Such materials have been used in thermal mapping and analysis in medical, industrial, and engineering applications such as dynamic heat transfer measurements [11,12,13,14,15,16,17]. 

Electrospinning has proven to be a useful approach to incorporate functional additives into fibers to fabricate functional materials [4]. On a bench scale, it is a simple, cost effective, small foot print, and versatile fiber spinning technology [1]. To generate nanofibers by electrospinning, an electric potential is applied between a capillary containing a polymer solution or melt and a grounded collector. The applied electric field leads to free charge accumulation at the liquid-air interface and electrostatic stress. When the electrostatic stress overcomes surface tension, the free surface deforms into a “Taylor cone”. Balancing the applied flow rate and voltage results in a continuous fluid jet from the tip of the cone. As the jet travels to the collector, it typically undergoes non-axisymmetric instabilities such as bending and branching leading to extreme stretching. As the fluid jet is stretched, the solvent rapidly evaporates to form the polymer fibers that are deposited onto a grounded target [18,19,20,21,22,23]. As a complex electrohydrodynamic process, the final fiber and mat/membrane properties depend on process parameters, set-up parameters, and solution properties. Recently, dispersions of cholesteric liquid crystal mixtures have also been blended with polyvinylpyrrolidone and electrospun [4,24]. There have been multiple step methods in which the liquid crystal was formulated, dispersed and the droplets were processed into fibers. The liquid crystal appeared to retain its thermochromic properties once processed [4,24].

Using a single fiber processing step, hierarchically structured nanofibers that incorporated functional materials have been achieved via coaxial electrospinning. In coaxial electrospinning, an electrospinnable polymer sheath solution is co-extruded with a desired core material through a spinneret of two coaxial capillaries. Under steady operation, continuous, coaxial streams of both core and shell fluids is observed as they exit the nozzle. The core fluid may be electrospinnable or may be one that does not readily form fibers when electrospun alone. The shell fluid envelops the core fluid and prevents the core fluid from breaking up into droplets [25,26]. 

Liquid crystals have been incorporated into the core of core-shell fibers via coaxial electrospinning. Nematic single component N-(4-methoxybenzylidene)-4-butylaniline (MBBA) [3] and multicomponent mixtures [7] were encapsulated within polyvinylpyrrolidone (PVP) based shells. Lagerwall et al. demonstrated that non-beaded fiber morphology was required for orientation of the liquid crystal and that confinement within the polymer stabilized the nematic state resulting in an increase in the clearing temperature [7]. Using a dual-core spinneret design, two nematic liquid crystals with different clearing points were encapsulated in adjacent cores [1]. Short pitch cholesteric liquid crystals have also been encapsulated by coaxial electrospinning; the fibers were spun from the LC in the cholesteric phase using a PVP based shell [6,27]. The resulting fibers demonstrated temperature sensitive iridescence [27].

One possible disadvantage is the complex set-up required i.e., multiple pumps, complex spinneret configurations [28]. Thus, scalability has also been a concern [29]. An alternative approach has been to electrospin a blend of a liquid crystal and polymer from a common solvent using a single spinneret [10]. West and co-workers electrospun polylactic acid (PLA) and 5-pentyl-4’-cyanobiphenyl (5CB) from a chloroform/acetone mixture. As the solvent evaporated, the 5CB phase separated and self-assembled forming a nematic core within a PLA shell [30]. The solvent induced phase separation process was also used to achieve PVP/5CB fibers from ethanol or a mixture of methanol and acetone [31].

Incorporating cholesteryl ester thermochromic liquid crystals using these exciting single step fiber processing approaches (coaxial and in situ solvent induced phase separation) has not been reported. Ability to use solvent induced phase separation with cholesteric liquid crystal mixtures is especially untested. Additionally, the results from the various methods have not been compared.

We aim to compare single step fiber spinning methods (i.e., blend electrospinning and coaxial electrospinning) and determine if it is possible to achieve thermochromic fibers by incorporating cholesteryl ester liquid crystal mixtures using these methods. We build on the existing literature of coaxial electrospinning liquid crystal (core) with a PVP/ethanol (shell), which has resulted in fibers with temperature sensitive optical properties [27]. However, PVP/ethanol was not an appropriate system for blend electrospinning because the cholesteryl ester liquid crystals are not fully miscible. As an alternative, we selected polystyrene (PS) which can be electrospun from toluene/acetone mixtures for blend electrospinning [32]. For coaxially electrospinning, we selected the same solvent system for the liquid crystal and co-extruded liquid crystal in toluene/acetone in the core with PVP/ethanol as the electrospinnable shell.

In this work, we compare blend electrospinning with coaxial electrospinning using the same solvent system for the liquid crystal in terms of fiber size, production rate, and liquid crystal loading within the fiber. The thermochromic properties of the resulting fibers using both methods are also examined. Specifically, we perform proof-of-concept experiments to demonstrate if the thermochromic properties of the liquid crystal are maintained during fiber processing comparing blend electrospinning and coaxial electrospinning.

## 2. Materials and Methods

### 2.1. Materials

Polystyrene (PS) (weight average molecular weight 350,000 g/mol) and polyvinylpyrrolidone (PVP) (weight average molecular weight 1,300,000 g/mol) were received from Sigma Aldrich (St. Louis, MO, USA). Toluene (ACS reagent grade), acetone (ACS reagent grade), and ethanol (Molecular Biology Grade) were received from Fisher Scientific (St. Louis, MO, USA). Liquid crystal components: cholesteryl oleyl carbonate (COC), cholesteryl benzoate (CB), and cholesteryl pelargonate (CP) were obtained from Sigma (St. Louis, MO, USA). All chemicals were used as received.

### 2.2. Formulation

#### 2.2.1. Liquid Crystal Formulation

To achieve thermochromic liquid crystals, we formulated ternary mixtures of cholesteryl benzoate (CB), cholesteryl pelargonate (CP), and cholesteryl oleyl carbonate (COC). The ternary mixtures were combined into homogenous solutions by combining appropriate masses of the three components (Table 1) at room temperature. The components were heated in an oil bath at 80–90 °C for 10 min, mixed by hand at room temperature for 5 min, and reheated in the oil bath at 80–90 °C for 10 min to fully melt the components. The resulting blends were clear and macroscopically homogenous. The liquid crystal was then cooled to room temperature before further use. 

#### 2.2.2. Polymer Solution Preparation

Various amounts (5–45 wt.%) of PS were dissolved in a mixture of toluene and acetone (7:3 v:v toluene: acetone) by stirring at ~250 rpm at room temperature overnight (at least 16 h) with a magnetic stirrer until macroscopically homogenous. Similarly, PVP (5–30 wt.%) was dissolved in ethanol by stirring at ~250 rpm at room temperature overnight (at least 16 h) with a magnetic stirrer until macroscopically homogenous. All solutions were stored at 4°C before further use. 

#### 2.2.3. Electrospinning Blend Preparation

PS solution and LC formulations were combined in various amounts with additional solvent (a mixture of toluene and acetone (7:3 v:v toluene: acetone)). The PS was held constant at 20 wt.%; the amount of LC was varied increased from 0 to 30 wt.%. The blends were stirred for ~1 hr at ~250 rpm at room temperature with a magnetic stirrer until macroscopically homogenous. All blends were stored at 4 °C before further use. 

### 2.3. Fiber Spinning

#### 2.3.1. Blend Electrospinning

For electrospinning, we used a point-plate configuration, where the PS/LC blend was loaded into a syringe fitted with a stainless-steel needle (0.508 mm I.D.) and attached to a power supply (Matsusada Precision Inc., model AU-40R0.75 with positive polarity, Kusatsu, Shiga, Japan). A flow rate of 0.5 mL/h (New Era Pump Systems syringe pump NE-300, Farmingdale, NY, USA), collecting distance of 10 cm between the tip of the needle and the ground collector plate covered with foil and applied voltage (positive polarity) of 5–12 kV were used. Blends were prepared and electrospun at temperatures between 19 and 22 °C and relative humidity between 11 and 20%.

#### 2.3.2. Coaxial Electrospinning

Coaxial electrospinning was performed with a custom-built apparatus comprised of two syringe pumps (New Era Pump Systems syringe pump NE-300, Farmingdale, NY, USA), and a custom coaxial spinneret (Rame Hart Coaxial 1418, Succasunna, NJ, USA; inner needle I.D./O.D. 0.033/0.049 and outer needle I.D./O.D. 0.063/0.083 inches). Similar to blend electrospinning, we used a horizontal point-plate configuration. The fibers were collected on a grounded plate covered in foil. The spinneret tip was attached to the high voltage power supply (Matsusada Precision Inc., model AU-40R0.75 with positive polarity, Kusatsu, Shiga, Japan). The PVP in ethanol solution was used as the shell solution. The core was LC dissolved in 7:3 v:v toluene:acetone to facilitate extrusion of the LC formulations at room temperature. The ratio of LC to solvent was varied. The flow rates of the core and shell were varied between 0.1 and 1.2 mL/h. Typically, the spinneret to collector distance was collecting 10 cm and the applied was 6–10 kV. For coaxial electrospinning, the materials were prepared and electrospun temperatures between 19 and 22 °C, and relative humidities between 45 and 55%.

### 2.4. Characterization

#### 2.4.1. Solution Characterization

The zero-shear viscosity of polymer solutions was measured at 25 °C using TA Instruments DHR-3 rheometer (New Castle, DE, USA) fitted with a 40 mm, 2° cone and plate geometry. To ensure uniform solution conditioning, a preshear was applied at a shear stress of 10 Pa for 30 s followed by a rest period of 120 s. The surface tension was measured using the pendant drop method [33,34] using a Rame-Hart Model 250 Goniometer with DROPimage software (Rame-hart, Succasunna, NJ, USA, USA). The method for surface tension was calibrated with ethanol. The conductivity of the solutions was measured using with an Oakton CON 150 conductivity meter from Cole-Parmer (Vernon Hills, Il USA 60061) and calibrated with a NIST traceable reference material (~10 Microseimen/cm, Webster, TX USA 77598). For the solution properties, the average and standard deviation of three measurements are reported. 

#### 2.4.2. Fiber Characterization

The structure of the fibers was investigated with polarized optical light microscopy (PLM) Eclipse 150N (Nikon Instruments Inc., Melville, NY, USA) with crossed polarizers (Epi Rotatable Polarizer and L-AN analyzer with CFI60-2 TU Plan FLUOR BD Objective Lenses). For higher resolution imaging, the fiber samples were analyzed with Scanning Electron Microscopy (Phenom ProX Desktop SEM, ThermoFisher Scientific, Waltham, MA, USA) with 10 kV accelerating voltage, and 15 mm working distance. Prior to analysis, the samples were sputter coated with gold (Polaron E 5100 (Quorum Technologies, Laughton, East Sussex, UK), 90 s, 1.5 kV, 10 mA, 10 Pa). The average fiber size was determined from at least 50 measurements using ImageJ software (US NIH, Bethesda, MD, USA). 

The thermochromic properties of the fiber were investigated using a Keyence VHX 5000 digital microscope (Keyence Corporation of America, Itasca, IL, USA) using a High-Resolution Zoom lens (VH-Z500T). The field diagram was adjusted to the “Normal” setting to obtain optimal balance between the outline of the fibers at their core. Illumination was set to full coaxial mode and the white balance was adjusted before imaging using PLM with crossed polarizers (OP-51649, Keyence, Itasca, IL, USA). Imaging of both blended and coaxial electrospun fibers was done at magnifications of 500×, 1000×, and 2000×. Changes of color as a function of temperature were captured using a heated stage and temperature controller (TC-1-100s, Microscope World, Carlsbad, CA, USA) operated between 18 °C and 40 °C and using 20%–25% DC. Samples were observed during both heating and cooling. 

## 3. Results

### 3.1. Blend Fiber Spinning—Polystyrene (PS)/LC Fibers

#### 3.1.1. Polystyrene/LC Blend Preparation

Our first step was to determine the compositions that formed single-phase blends and did not phase separate into a polymer-rich and solvent-rich phase. Phase separation was observed when the liquid crystal concentration was greater than 20 wt.%. Thus, subsequent experiments were performed with 20 wt.% PS and a maximum of 20 wt.% LC.

#### 3.1.2. Effect of LC Concentration

We investigated the effect of LC concentration using LC-3 on fiber formation. Notably 20 wt.% PS without LC forms beaded fibers (Figure 1A). Introducing small amounts of LC (5% LC loading) increases the amount of beading (Figure 1B_. Fibers with 10% and 15% LC loading also contain some beading (Figure 1C and Figure 1D). The number of beads is reduced compared to 5% LC loading. Interestingly, increasing the LC loading to 20% LC loading suppresses bead defects and uniform fibers are achieved (Figure 1E). 

Typically, when electrospinning a blend system, one component is the “carrier” polymer and forms uniform fibers alone. This result is surprising because neither PS nor LC alone forms uniform fibers and serves as the “carrier” polymer in this system. 

#### 3.1.3. Effect of LC Formulation

We examined the effect of LC formulation using blends of 20 wt.% PS and 20 wt.% LC. Figure 2A shows 20 wt% PS only for comparison, which forms beaded fibers. Interestingly, at the same concentration, LC-3 seemed to form uniform fibers (Figure 2D) whereas LC-1 had bead defects (Figure 2B). LC-2 seemed to form uniform fibers with minimal bead defects (Figure 2C). This result suggests that increasing the CP content in LC formulation and the resulting blend promotes fiber uniformity. 

Further examining the effect of LC formulation on ability to form uniform fibers, we performed fibers size analysis with at least 50 measurements. The fiber size distributions are shown in Figure 3. For the 60:30:10 formulation, the fiber size diameter had a wide range from 20 to 70 μm with a multimodal distribution. Interestingly, increasing the CP content to the 45:45:10 formulation, the fiber size diameter size was between 10 and 30 microns with the greatest number of fibers around 20 μm. Further increasing the CP content to the 30:60:10 formulation again resulted in a wide range of fiber diameters from 5 to 70 μm with a multimodal distribution. We quantified the uniformity by examining the coefficient of variation of the fiber diameter (Table 2). We note that introducing the liquid crystal significantly increased the fibers size compared to polystyrene only fibers from approximately 5 μm to 20–40 μm (Table 2). Increasing the solids content in the blend is expected to increase the fiber diameter. The beaded polystyrene only fibers resulted in a relatively large coefficient of variation of the fiber diameter of 56% (Table 2). Generally, introducing the liquid crystal formulations improved the uniformity as indicated by the decrease in the coefficient of variation of the fiber diameter (Table 2). Interestingly, despite a few bead defects evident in Figure 2, the most uniform fibers, statistically when analyzed over multiple images, were 20 wt.% PS, 20 wt.% LC-2 (45:45:10) as indicated be the lowest coefficient of variation (less than 20%) (Table 2). 

#### 3.1.4. Blend Solution Properties

To further investigate the system, we measured the blend solution properties that primarily affect ability to form fibers, namely viscosity, surface tension, and conductivity (Table 3). We note none of the electrospinning blends showed measurable conductivities. Interestingly, LC-1 and LC-2 appeared to have slightly increased viscosity and surface tension compared to 20 wt.% PS and LC-3. However, such trends would not favor uniform fiber formation. The blend with LC-3 shows lower viscosity and lower surface tension compared to LC-1 and LC-2 at the same concentration. This result suggests that CP acts as a plasticizer in this system. 

The viscosity and surface tension of 20 wt.% PS with 20 wt.% LC-3 were comparable to 20 wt.% PS without LC. Despite comparable solution properties, the blend containing LC-3 forms uniform fibers whereas PS alone forms beaded fibers. Ability to form fibers upon the addition of LC-3 cannot be explained by a change in solution properties.

In this case, fiber formation may be attributed to strong intermolecular interactions leading to supramolecular assemblies. Systems with strong intermolecular interactions such as trisamides and modified cyclodextrins that form supramolecular assemblies that aggregate can form fibers in the presence of small amounts of polymer or in the absence of polymer [35,36,37]. At increased LC concentrations, LC-LC interactions may promote polymer-polymer interactions which increase the effective entanglements and facilitate uniform fiber formation. 

### 3.2. Coaxial Fiber Spinning—Polyvinylpyrrolidone (PVP)/LC Fibers

Since liquid crystals alone could not be electrospun into fibers (Appendix A), we also explored coaxial electrospinning as a means to produce liquid crystal containing fibers. In coaxial electrospinning, two solutions are electrospun through a spinneret of two coaxial nozzles. Generally, the electrospinning solution for the shell must be electrospinnable. The core fluid does not have to readily form fibers when electrospun alone. Thus, coaxial extrusion enables electrospinning of fluids that are difficult to process. The shell fluid envelops the core fluid and prevents the core fluid from breaking up into droplets. Stabilization occurs due to viscoelasticity of the shell solution and reduced surface tension at the core-shell interface. Selecting a common solvent results in particularly low interfacial tension. Since electrospinning is a relatively fast process, the core and shell solutions may or may not be miscible; the two fluids do not mix significantly over the short duration of the electrospinning process. One important consideration is solubility of the polymer solutions, i.e., the polymers must not precipitate at the fluid interface [25,26].

Analogous to conventional electrospinning, solution properties of both solutions and process parameters e.g., applied electric field strength, flow rates, etc. affect fiber quality. The flow rates of the core and shell solutions is of particular practical importance. If the flow rate of the core is too high, the core fluid breaks up into droplets. If the flow rate of the shell is too high, the spinning of the core is not continuous and the fibers form without a continuous thread of core material. Generally, having the core flow rate lower than the sheath flow rates promotes stable jetting of both fluids. Further, the ratio of the diameter of the core to the shell and core loading are dictated by the flow rates [25,26].

To produce LC containing fibers via coaxial electrospinning, we selected polyvinylpyrrolidone (PVP) as the shell polymer because it is electrospinnable. Additionally, it has been previously used for coaxially electrospinning with liquid crystals and the resulting fibers have had temperature sensitive optical properties [27].We selected ethanol as the solvent because it is compatible with the LC components i.e., should not cause the LC components to precipitate. To extrude the LC, we included small amounts of solvent (the 7:3 ratio of toluene:acetone used for blend electrospinning). 

First, we determined the PVP concentration in ethanol required to form uniform fibers. We examined the effect of LC:solvent ratio on the resulting fibers. Then, we examined the effect of process parameters on fiber structure. Based on literature [25,26], we focused on the core:shell flow rate ratio and total flow rate. Since LC-2 produced the most uniform fibers with blend electrospinning as quantified by the lowest coefficient of variation in the fiber diameter (Table 2), LC-2 was used in these coaxial experiments as a model system since our focus was comparing the fiber preparation methods. 

#### 3.2.1. PVP Electrospinning

We began by determining the PVP concentration required to electrospin uniform fibers. We varied the concentration from 5 wt.% to 25 wt.% PVP in ethanol. We observed that at low PVP concentrations e.g., 5 wt.% beads on a string morphologies were observed (Figure 4A). Increasing the PVP concentration to 10 wt.% a transition to beaded fibers was observed (Figure 4B) indicating jet break-up. With increasing polymer concentration, there was a transition to uniform fibers; 15-20 wt.% PVP solutions produced relatively uniform fibers (Figure 4C and Figure 4D). Interestingly, with further increases in polymer concentration, we observed formation of fused fiber bundles (Figure 4E). These observations are similar to other electrospinnable polymer systems in which at low viscosities/polymer concentrations, the electropinning jet breaks up into droplets rather than stretching to form a fiber. With increasing concentration, there is a transition to beaded fibers and a second transition to uniform fibers. The ability to form uniform fibers has been frequently attributed to polymer entanglement [38,39,40]. 

#### 3.2.2. Coaxial Electrospinning

Moving forward, coaxial spinning was performed with 15 wt.% PVP in ethanol. In order to co-extrude the LC with the PVP solutions at room temperature, small amounts of solvent toluene:acetone (7:3 v:v) were added to create the core solution. We first examined the effect of the amount of solvent mixture added to the LC in the core solution. We increased the amount of solvent from 15:1 to 5:1 LC to solvent (by mass). We held the core flow rate ratio constant at 0.1 mL/hr. The shell flow rate was 0.2 mL/hr or 0.3 mL/hr. At a 15:1 LC:solvent ratio, there did not seem to be sufficient solvent to continuously extrude the LC and large globs of LC were observed (Figure 5A and Figure 5B). At a 5:1 LC:solvent ratio, the LC appeared to form droplets within the fiber (Figure 5E and Figure 5F). At an intermediate ratio of 10:1 LC:solvent, incorporating the LC seemed to have minimal effect on the PVP shell (Figure 5C and Figure 5D). Therefore, moving forward, a core of 10:1 LC:solvent was used.

Our goal was to produce uniform fibers. Thus, we examined the effect of electrospinning process parameters on fiber structure. Our focus was total flow rate which dictates throughput and flow rate ratio which dictates LC loading in the fiber. To achieve relatively high LC loading in the fiber, we initially worked at a 2:1 shell to core ratio. Globs of LC were observed at the lowest flow rate (0.2 mL/hr shell: 0.1 mL/hr core). These defects increased as the total flow rate increase at a constant 2:1 shell to core ratio (Appendix A). To increase fiber uniformity, we increased the flow rate ratio of the shell to the core. By increasing the shell flow rate:core flow rate ratio to 4:1, uniform fibers were achieved at throughputs of 0.5 mL/hr and 1 mL/hr, with some bead defects apparent at 1.5 mL/hr (Appendix A).

We further characterized the structure of the coaxially electrospun fibers using SEM (Figure 6). The coaxially electrospun fibers that incorporating LC into the core (Figure 6B) are compared to PVP coaxially electrospun with a solvent core (Figure 6A) for comparison. The fibers incorporating the LC are much larger than the fibers with the solvent only core. The average fiber size with solvent only is 5.3 ± 2.7 μm. For the fibers with liquid crystal the average fiber diameter 23 ± 20. μm. The increase in fiber diameter suggests successful incorporating of the LC during coaxial electrospinning. We note there is also a decrease in fiber uniformity upon incorporating the LC as indicated by the increase in the coefficient of variation from 51% to 87%. This result may be due to a combination of empty PVP fibers (~5 μm) and LC-loaded fibers (~25 μm) (Figure 6C). Overall, with coaxial electrospinning, we estimate the mass throughput of fibers is 0.3-0.4 g/hr and the LC loading is between 50 and 60% LC loading.

### 3.3. Thermochromic Fiber Characterization

After preparing fibers using blend electrospinning and coaxial electrospinning using the same solvent system for the liquid crystal, we next aimed to perform proof-of-concept experiments to demonstrate if the thermochromic properties of the liquid crystal were maintained during fiber processing.

#### 3.3.1. Blend Fibers

Polarized light microscopy was performed on the 20 wt.% PS, 20 wt.% LC-2 fibers and compared to 20 wt.% PS only fibers (Figure 7). Portions of the fiber containing LC-2 appear blue which are not apparent in the PS only fiber suggesting the successful incorporation of the liquid crystal as the mesophase transition temperature associated with the thermochromic behavior is reported to be around 27 °C [41]. However, no color change was observed upon heating the stage to 50 °C; the liquid crystal is expected to change color between 26–30 °C [41]. We note, upon heating the stage from room temperature to ~50 °C, there was evidence of a phase change in the fibers containing LC representative data from fibers containing LC-1 are shown in Appendix A whereas no change was observed in the PS only fibers (Appendix A). This result suggests further suggests that the liquid crystal is successfully incorporated into the blend fiber, but does not retain its thermochromic properties once processed into the fiber. Blend processing with polystyrene using toluene/acetone may affect liquid crystal orientation in the fiber, consistent with previous reports [7].

#### 3.3.2. Coaxial Fibers

Coaxially electrospun fibers containing the LC-2 were heated and observed under PLM. Representative images demonstrating the observed thermochromic behavior in the fibers containing. Specifically, the LC in the core was observed to transition from red to blue as the fibers were heated as shown in Figure 8 whereas no change was observed in the PVP shell:solvent core fibers (Appendix A). This result confirms that the LC was successfully incorporated into the fiber during coaxial electrospinning. Furthermore, the incorporated LC retains its thermochromic properties such that the fiber demonstrates thermochromic behavior. We note that cooling of the fiber from 35 °C to 20 °C resulted in a transition from blue to red, demonstrating the reversible thermochromic properties of the LC mixture within the coaxially electrospun fiber and multiple heating cycles were performed demonstrating a reversible phase transition.

These proof-of-concept experiments demonstrate that the thermochromic properties of the liquid crystal can be retained upon coaxial electrospinning to achieve thermochromic fibers. In the future, the thermochromic behavior of the three LC formulations upon heat and cooling at various heating rates should be quantified. Furthermore, this technique can be expanded to other liquid crystal formulations with tunable transition temperatures for the desired application.

### 3.4. Fiber Spinning Method Comparison

A table summarizing the results of the fibers obtained with blend electrospinning with polystyrene from a 7:3 v:v mixture of toluene:acetone and coaxial electrospinning with a shell of 15 wt.% PVP and a core of 10:1 LC:solvent by weight (with 7:3 v:v acetone: toluene) is shown in Table 4.

Coaxial electrospinning produces fibers with higher nominal fiber production rates (g/hr) and with higher nominal LC content in the fiber (wt. LC/wt. polymer assuming all of the solvent evaporates) but larger fiber size distributions than blend electrospinning with a single nozzle. Importantly, coaxially electrospinning preserves the thermochromic properties of the LC so that the fibers have thermochromic properties.

In this work, we have compared coaxial electospinning and blend electrospinning demonstrating proof-of-principle that the thermochromic properties of the LC are maintained when coaxial electrospinning using PVP as the shell polymer. Examining the effect of the properties of the shell polymer on the resulting fiber (e.g., refractive index, crystallinity, etc.) will be pursued in future work. 

## 4. Conclusions

In this work, we have incorporated cholesteric liquid crystals into polymer fibers via electrospinning using two methods: (1) blend electrospinning and (2) coaxial electrospinning. For blend electrospinning, we determined the concentration of LC that can be incorporated into a polystyrene solution using a 7:3 v:v toluene:acetone mixture as a solvent prior to phase separation into a polymer rich phase and a solvent rich phase. Since bead formation can be suppressed by the addition of 20 wt.% PS, intermolecular interactions seem to be important in facilitating fiber formation in the polystyrene-LC blend systems. Building on existing literature coaxially electrospinning LC, we have coaxially electrospun liquid crystals with polyvinylpyrrolidone (PVP). Specifically, 15 wt.% PVP in ethanol was used as the shell solution, and a 10:1 LC:solvent was used as the core. The solvent for the LC core was 7:3 v:v toluene:acetone. A 4:1 shell:core flow rate ratio led to uniform fibers with highest throughputs. Comparing the two electrospinning methods, coaxial electrospinning produces fibers with higher nominal fiber production rates (g/hr) and with higher nominal LC content in the fiber (wt. LC/wt. polymer assuming all of the solvent evaporates) but larger fiber size distributions than blend electrospinning with a single nozzle. Notably, our proof-of-concept experiments demonstrate that coaxially electrospinning preserves the thermochromic properties of the LC so that thermochromic fibers are achieved.

## Figures and Tables

**Figure 1 polymers-12-00842-f001:**
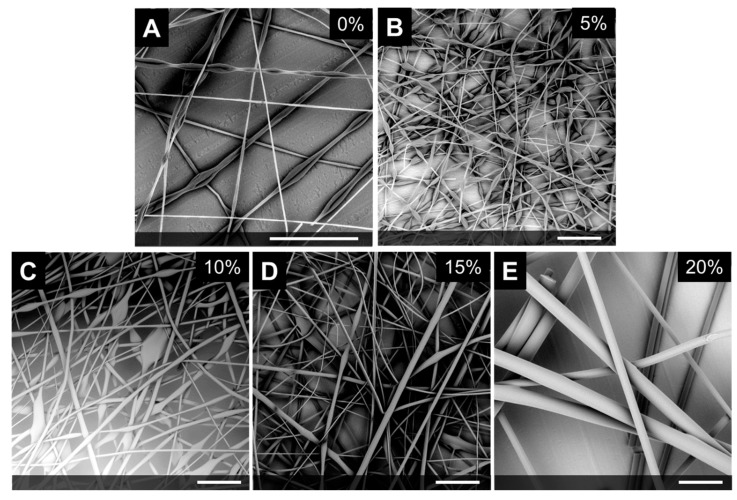
SEM micrographs of 20 wt% PS with increasing amounts of LC using LC -3. The scale bar included is 100 microns in length. Interestingly, (**A**) shows 20 wt.% PS (without LC) forms beaded fibers). Adding small amounts of LC (5 wt.% shown in (**B**), 10 wt.% shown in (**C**), or 15% wt.% in solution shown in (**D**)) also produces beaded fibers. A 20 wt.% PS, 20 wt.% LC blend shown in (**E**) forms uniform fibers.

**Figure 2 polymers-12-00842-f002:**
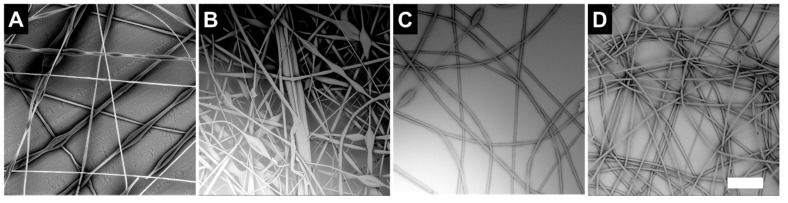
SEM micrographs showing the effect of LC formulation on fiber structure. Fibers spun from (**A**) 20 wt.% PS (**B**) 20 wt.% PS, 20 wt.% LC-1, (**C**) 20 wt.% PS, 20% LC -2, and (**D**) 20 wt.% PS, 20 wt.% LC-3 with a 100 micron scale bar.

**Figure 3 polymers-12-00842-f003:**
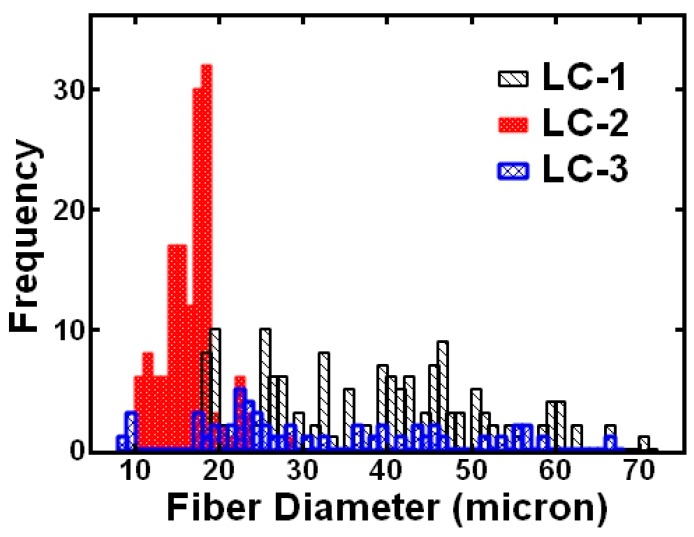
Fiber size distributions for 20 wt.% PS, 20 wt.% LC for the three LC formulations.

**Figure 4 polymers-12-00842-f004:**
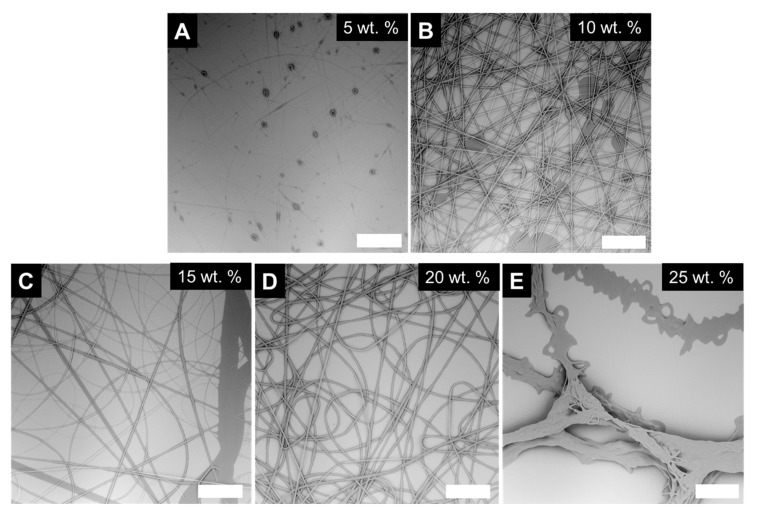
SEM micrographs of polyvinylpyrrolidone (PVP) electrospun from increasing concentrations in ethanol. (**A**) Low concentrations (5 wt.% PVP) produced bead on a string morphologies, (**B**) 10 wt.% produced beaded fibers, (**C**) 15 wt.%, and (**D**) 20 wt.% produced relatively uniform fibers, and (**E**) high concentrations (25 wt.%) produced fused fiber bundles. The scale bar is 100 microns in length.

**Figure 5 polymers-12-00842-f005:**
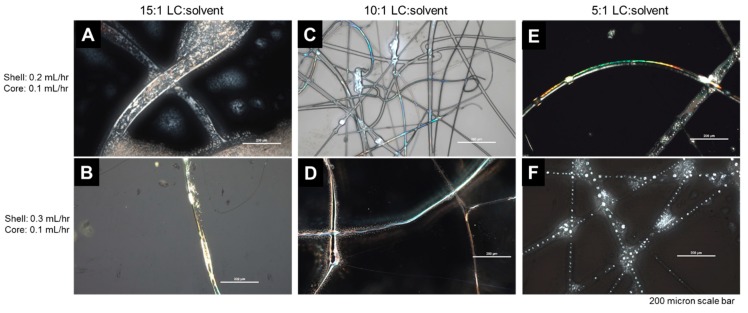
Polarized optical light microscopy (PLM) images of PVP fibers electrospun with various LC:solvent cores. The core flow rate was held constant at 0.1 mL/h: the shell flow rate was 0.2 or 0.3 mL/hr. (**A**) and (**B**) At a 15:1 LC:solvent ratio, large globs of LC were observed. (**C**) and (**D**) At an intermediate ratio of 10:1 LC:solvent, incorporating the LC appeared to have minimal effect on the PVP shell, while facilitating continuous extrusion of the LC. (**E**) and (**F**) At a ratio of 5:1 LC:solvent, the LC appeared to form droplets within the fiber. Increasing the shell to core flow rate ratio generally improved fiber formation. Therefore, we worked with a 10:1 LC:solvent mass ratio using a 7:3 v:v toluene:acetone mixture.

**Figure 6 polymers-12-00842-f006:**
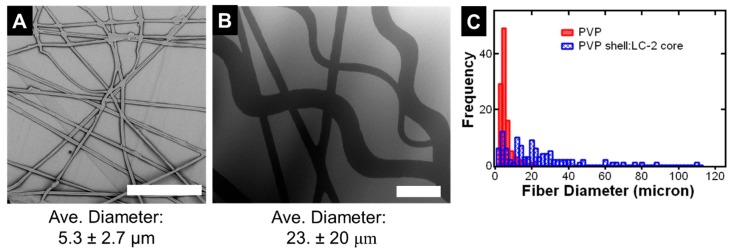
SEM micrographs of coaxially spun 15 wt. % PVP in ethanol shell (0.8 mL/hr) with (**A**) a 7:3 v:v toluene:acetone core (0.2 mL/hr) compared to (**B**) a 10:1 LC-2:solvent core both with a 100 μm scale bar. Increase of the fiber diameter suggests successful incorporation of the LC. (**C**) Fiber size distributions comparing sample (A) and (B).

**Figure 7 polymers-12-00842-f007:**
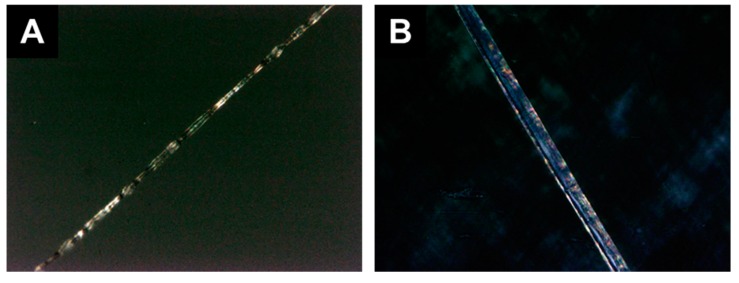
Polarized light microscopy of (**A**) 20 wt.% PS and (**B**) 20 wt.% PS with 20 wt.% LC-2 fibers at 25 °C. Portions of the fiber containing LC-2 appear blue which are not apparent in the PS only fiber indicating the successful incorporation of the liquid crystal. However, no color change was observed upon heating to 50 °C.

**Figure 8 polymers-12-00842-f008:**
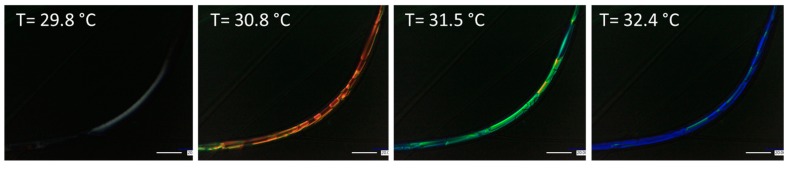
Polarized light microscopy of coaxially electrospun PVP shell, LC-2 core fibers heated from ambient conditions. Thermochromic behavior, i.e., a transition from red to blue, was observed upon heating. The temperatures indicated are the stage temperature.

**Table 1 polymers-12-00842-t001:** Compositions of thermochromic liquid crystal (LC) formulations which are ternary mixtures of cholesteryl benzoate (CB), cholesteryl pelargonate (CP), and cholesteryl oleyl carbonate (COC).

Sample	COC (wt.%)	CP (wt.%)	CB (wt.%)
LC-1	60	30	10
LC-2	45	45	10
LC-3	30	60	10

**Table 2 polymers-12-00842-t002:** Average fiber sizes for 20 wt.% PS, 20 wt.% LC for the three LC formulations.

Sample	Avg. Fiber Diameter (μm)	Std. Dev. (μm)	Coefficient of Variation (%)
20 wt.% PS	4.8	2.6	53
20 wt.% PS, 20 wt.% LC-1	38	13	34
20 wt.% PS, 20 wt.% LC-2	17	3	18
20 wt.% PS, 20 wt.% LC-3	32	15	47

**Table 3 polymers-12-00842-t003:** The 20 wt.% PS, 20 wt.% LC blend solution properties for three LC formulations.

Sample	Viscosity (Pa*s)	Surface Tension (mN/m)	Conductivity (μS/cm)
20 wt.% PS	1.1 ± 0.3	23 ± 1	N.D.
20 wt.% PS, 20 wt.% LC-1	1.3 ± 0.1	31 ± 2	N.D.
20 wt.% PS, 20 wt.% LC-2	1.7 ± 0.1	33 ± 2	N.D.
20 wt.% PS, 20 wt.% LC-3	1.0 ± 0.2	22 ± 2	N.D.

**Table 4 polymers-12-00842-t004:** Summary of fibers produced from blend electrospinning and coaxial electrospinning using LC-2.

	Blend	Coaxial
Fiber Diameter	17 ± 3 μm	23 ± 20 μm
Nominal Fiber Throughput (g/hr)	0.2	0.5
Nominal LC loading (wt. LC/wt. polymer)	50%	60%
Thermochromic behavior	No	Yes

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
