# Peer review of "Thermochromic Fibers via Electrospinning"

_polymers, 2020, doi:10.3390/polym12040842_

Round 1

Reviewer 1 Report

The paper entitled ‘Thermochromic Fibers via Electrospinning’ is a study aiming to achieve thermochromic fibers by incorporating the liquid crystal (LC) formulations into fibers by blend and co-axial electrospinning. The authors show rather preliminary results that probably need to do more characterization of obtained fibers in the following up research. Therefore, before the recommendation to publish this work, I would like to ask the authors for a few clarifications on the following points:

  1. Specify what is the room temperature, add the relative humidity during all experiments also during preparation of solutions for electrospinning
  2. The stirring conditions should be mentioned, especially rotation speed and time
  3. For SEM imaging the working distance is missing and the information about the coating prior imaging
  4. Adding a) b) c) markers and related descriptions will help to read figure caption and related figures to the discussion points.
  5. Why co-axial electrospinning was not performed with PS too?
  6. You are comparing the output of electrospinning with two different polymers PVP in co-axial and PS in a blend? If you do not use the same polymer for both methods you cannot compare the output of the process.

Reviewer 2 Report

In this paper, the author incorporated cholesteric liquid crystals into polymer fibers via electrospinning using two methods (1) blend electrospinning and (2) coaxial electrospinning. And the coaxially electrospinning preserves the thermochromic properties of the LC so that thermochromic fibers are achieved. The article is well written, but there are still some issues, so we recommend to reconsider whether to accept after major revision. Some questions and suggestions are shown below.

1、In Figure 2, each figure should have a scale. Moreover, the scales of Figure 1 and Figure 2 are preferably the same, so that the price comparison can be performed more intuitively.

2、The fiber size distributions of Figures 3 and 6 are not intuitive enough.

3、From Figure 2, we can see that LC-2 fiber still has a beaded structure. So why use LC-2 instead of LC-3 for core-shell electrospinning?

4、In Figure 7, why use LC-1 fiber for thermochromic test instead of LC-2 or LC-3 fiber?

5、What is the temperature selection basis for thermochromic testing? Why is the temperature range set at 20-35 ° C? It can be seen from Figure 8 that the temperature difference of the color change is very small (308.-31.5-32.4 ℃), so in this case, is there any application prospect of this discoloration feature?

Round 2

Reviewer 1 Report

-

Reviewer 2 Report

In this paper, the author incorporated cholesteric liquid crystals into polymer fibers via electrospinning using two methods (1) blend electrospinning and (2) coaxial electrospinning. And the coaxially electrospinning preserves the thermochromic properties of the LC so that thermochromic fibers are achieved. The article is well written and it has been modified in accordance with comments. It is recommended to accept it.